# Target-Agnostic Gender-Aware Contrastive Learning for Mitigating Bias in Multilingual Machine Translation

**Minwoo Lee**[1*]  **Hyukhun Koh**[2]  **Kang-il Lee**[1]
**Dongdong Zhang**[3]  **Minsung Kim**[1]  **Kyomin Jung**[1,2]

[1]Dept. of ECE, Seoul National University,  [2]IPAI, Seoul National University,
[3]Microsoft Research Asia

{minwoolee, hyukhunkoh-ai, 4bkang, kms0805, kjung}@snu.ac.kr
dozhang@microsoft.com

## Abstract

Gender bias is a significant issue in machine translation, leading to ongoing research efforts in developing bias mitigation techniques. However, most works focus on debiasing bilingual models without much consideration for multilingual systems. In this paper, we specifically target the gender bias issue of multilingual machine translation models for unambiguous cases where there is a single correct translation, and propose a bias mitigation method based on a novel approach. Specifically, we propose **G**ender-**A**ware **C**ontrastive **L**earning, **GACL**, which encodes contextual gender information into the representations of non-explicit gender words. Our method is target language-agnostic and is applicable to pre-trained multilingual machine translation models via fine-tuning. Through multilingual evaluation, we show that our approach improves gender accuracy by a wide margin without hampering translation performance. We also observe that incorporated gender information transfers and benefits other target languages regarding gender accuracy. Finally, we demonstrate that our method is applicable and beneficial to models of various sizes.[1]

## 1 Introduction

In machine translation research, gender bias has emerged as a significant problem, with recent neural machine translation (NMT) systems exhibiting this bias (Prates et al., 2020). This bias can manifest in various ways, such as the misgendering of individuals based on stereotypes or defaulting to masculine gender translations. As a result, there is a growing need to address and mitigate gender bias in machine translation systems to ensure fair and unbiased translations that accurately reflect the intended meaning without perpetuating gender-based assumptions.

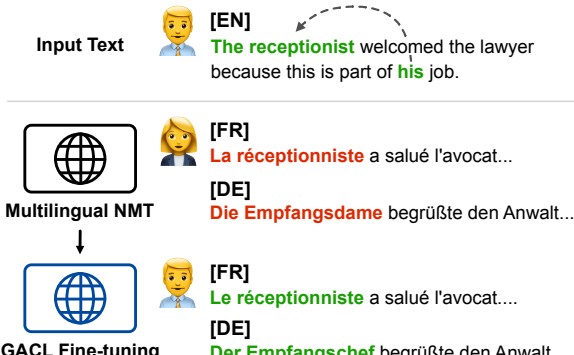

Figure 1: Example sentence from the WinoMT benchmark and the corresponding translation outputs of multilingual NMT systems. Existing systems often fail to translate with correct gender inflections.

Efforts have been made in recent studies to mitigate the gender bias issue in machine translation (Choubey et al., 2021, Saunders and Byrne, 2020, Costa-jussà and de Jorge, 2020). However, most of the works focus on mitigating bilingual NMT models and evaluate on a single language direction. Recently, Costa-jussà et al. (2022) demonstrated that the shared encoder-decoder architecture in multilingual NMT systems leads to worse gender accuracy compared to language-specific modules. Nonetheless, it remains unclear whether the existing debiasing methods would yield similar effectiveness in multilingual NMT models.

In this work, we investigate in detail the gender bias issue of multilingual NMT models. We focus on translating unambiguous cases where there is only one correct translation with respect to gender. We consider multiple target languages simultaneously with various gender-based metrics and find that even the state-of-the-art multilingual NMT systems still exhibit a tendency to prefer gender stereotypes in translation.

Therefore, we propose a new debiasing method for multilingual MT based on a new perspective of the problem. We hypothesize that the gender bias in unambiguous settings is due to the

---

*Work done during internship at MSRA.

[1]Code available at https://github.com/minwhoo/GACL

lack of gender information encoded into the non-explicit gender words and devise a scheme to inject correct gender information into their latent embeddings. Specifically, we develop **G**ender-**A**ware **C**ontrastive **L**earning, **GACL**, which assigns gender pseudo-labels to text and encodes gender-specific information into encoder text representations. Our method is agnostic to the target translation language as the learning is applied on the encoder side of the model and can be applied to debias pre-trained NMT models through fine-tuning. We also evaluate whether existing debiasing techniques for bilingual NMT are equally effective for multilingual systems and compare their effectiveness on different target languages.

Experimental results show that our method is highly effective at improving gender bias metrics for all 12 evaluated languages, with negligible impact on the actual translation performance. We find our approach applicable to various model architectures and very efficient in that it demonstrates significant gender accuracy improvement with just a few thousand steps of fine-tuning. We also discover that the debiasing effects extend to target language directions that are not trained on previous models. Through further analysis, we demonstrate that our method effectively incorporates contextual gender information into the model encoder representations.

In summary, the contributions of our work are as follows:

- We find that recent multilingual NMT models still suffer from gender bias and propose **GACL**, a novel gender debiasing technique for multilingual NMT models based on contrastive learning.

- To the best of our knowledge, we are the first to show that the gender debiasing effect transfers across other languages on multilingual NMT models that were not fine-tuned.

- Through extensive evaluation and analysis, we show that our method is effective across multiple architectures while having a negligible impact on translation performance.

## 2 Method

In this paper, we propose GACL, a gender-aware contrastive learning method for mitigating the unambiguous gender bias issue in multilingual machine translation. Our approach is applicable to

**1. Gender-related data filtering**

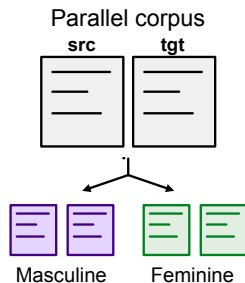

**2. Gender-aware Contrastive Learning**

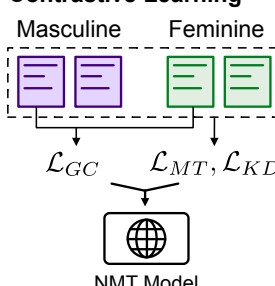

Figure 2: Overview of our proposed method. Train data is first filtered for sentences containing gender-related words. The processed dataset is used to fine-tune NMT model using machine translation and gender-aware contrastive learning objectives.

multilingual models that have encoder-decoder architectures and support English-to-many translation directions. We filter gender-related data and fine-tune the pre-trained NMT model with the filtered data. The overview of the method is shown in Figure 2.

### 2.1 Data Filtering and Preprocessing

We first filter the parallel train data for sentence pairs that contain gendered words in the English source sentence using the gender-related word list by Zhao et al. (2018). We exclude sentences that contain either no gendered words or gendered words of both male and female genders. After filtering the train data, we undersample the larger of the sentence pairs containing male and female gender words so that the number of samples for each gender is the same, in similar fashion to Choubey et al. (2021).

### 2.2 Gender-aware Contrastive Loss

We devise a contrastive loss that incorporates gender information into the encoder embeddings. Although the optimal approach would be to apply the contrastive scheme exclusively to words that exhibit gender-based translation variations, this varies depending on the translated language and is challenging to know in advance. Hence, we use mean-pooled sentence-level embeddings for our contrastive learning scheme instead.

Given $h_i$ as the encoder embedding of the source sentence, we define positive samples to be the set of sentence representations that have the same gender as $h_i$ and negative samples as the set of representations that have the opposite gender. We correspondingly formulate contrastive loss as follows:

$$\mathcal{L}_{GC}^{(i)} = - \sum_{h^+ \in H_i^+} \log \frac{e^{sim(h_i, h^+)/\tau}}{\sum_{h^* \in H_i^+ \cup H_i^-} e^{sim(h_i, h^*)/\tau}},$$

where $H_i^+$ is the set of positive samples, $H_i^-$ is the set of negative samples, $sim(\cdot, \cdot)$ is the cosine similarity function, and $\tau$ is the temperature hyperparameter. Our formulation is equivalent to the supervised contrastive loss by Khosla et al. (2020), where we use the gender information as pseudo-labels to define positive and negative pairs. In practice, we use positive samples $H_i^+ = \{h_i'\} \cup \{h_j | g_j = g_i\}$ where $h_i'$ is the representation based on different dropout seed, as in Gao et al. (2021), and $h_j$ are in-batch samples with the same gender marking $g_i$. For negative samples, we use $H_i^- = \{h_k | g_k \neq g_i\}$ where $h_k$ are the in-batch samples with different gender markings.

In addition to the gender-aware contrastive loss, we train our model with the original machine translation loss to prevent forgetting. We also add knowledge distillation loss with the frozen machine translation model as the teacher model to preserve the translation performance (Shao and Feng, 2022). In sum, our training objective is as follows:

$$\mathcal{L}_{train} = (1 - \alpha) \cdot \mathcal{L}_{MT} + \alpha \cdot \mathcal{L}_{KD} + \lambda \cdot \mathcal{L}_{GC},$$

where the machine tranlation loss $\mathcal{L}_{MT}$ and the knowledge distillation loss $\mathcal{L}_{KD}$ is added with weights based on hyperparameter $\alpha$, and our proposed loss $\mathcal{L}_{GC}$ is added with multiplied hyperparameter $\lambda$.

## 3 Experimental Framework

In this section, we describe the details of the experiments, including the data, metrics, baseline methods, training architecture, and parameters.

### 3.1 Dataset and Metrics

In order to measure the unambiguous gender bias in machine translation systems, we employ two evaluation benchmarks: WinoMT and MT-GenEval.

**WinoMT** (Stanovsky et al., 2019) is a widely used gender bias evaluation benchmark consisting of 3,888 English sentences, where each sentence contains an occupation and a gendered coreferential pronoun. WinoMT supports ten target languages: German, French, Italian, Ukrainian, Polish, Hebrew, Russian, Arabic, Spanish, and Czech.

Four metrics are used to measure the gender bias with the WinoMT dataset.

**Accuracy** measures whether the occupation is translated with the correct gender inflection based on the pronoun. The occupation word is determined using source-target alignment algorithm, and the inflected gender is detected using target language-specific morphological analysis.

$\mathbf{\Delta G} = Acc_{male} - Acc_{female}$ measures the difference in accuracy between male sentences and female sentences.

$\mathbf{\Delta S} = Acc_{pro} - Acc_{anti}$ measures the difference in accuracy between sentences with pro-stereotypical and anti-stereotypical gender-occupation pairings as defined by Zhao et al. (2018).

$\mathbf{\Delta R} = Recall_{male} - Recall_{female}$, suggested by Choubey et al. (2021), measures the difference in the recall rate of male and female sentences.

**MT-GenEval** (Currey et al., 2022) is a recently released gender accuracy evaluation benchmark that provides realistic, gender-balanced sentences in gender-unambiguous settings. We use the counterfactual subset, where for each sentence, there exists a counterfactual version in the set with only the gender changed. MT-GenEval supports eight target languages: Arabic, German, Spanish, French, Hindi, Italian, Portuguese, and Russian.

Four metrics are used to measure the gender bias with the MT-GenEval dataset.

**Accuracy** is measured based on whether the unambiguously gendered occupation has the correct gender inflection. Unlike WinoMT, however, accuracy is measured differently; using the counterfactual variants, words that are unique to a single gender are extracted for each sentence, and a translation is marked correct if the words unique to a different gender are not included in the translation.

However, this definition of accuracy has a problem in that even if the translation is incorrect, it could still be marked correct if the words unique to the different gender are not be contained. To avoid this problem, we define an alternative measure of accuracy, denoted **Explicit Accuracy**. In measuring explicit accuracy, a translation is marked correct if the words unique to different gender are not included in the translation, and the words unique to the same gender are *explicitly* included in the translation. This definition makes explicit accuracy a stricter version of the original accuracy.

$\mathbf{\Delta G} = Acc_{male} - Acc_{female}$ and $\mathbf{E\text{-}\Delta G} =$

$ExplicitAcc_{male} - ExplicitAcc_{female}$ measures the difference of male and female sentences in terms of accuracy and explicit accuracy respectively.

We use the **FLORES-200** (Costa-jussà et al., 2022), a standard multilingual NMT benchmark, to measure the translation performance of our models. FLORES-200 consists of 3,001 sentences sampled from English Wikimedia projects and professionally translated into 200+ languages. We use SentencePiece BLEU (spBLEU) and ChrF++ as evaluation metrics.

### 3.2 Baselines

We compare three baseline methods that have been previously proposed to mitigate gender bias in machine translation.

**Balanced:** Costa-jussà and de Jorge (2020) proposed to filter existing parallel corpora for sentences with gender mentions and subsample the data to create a balanced version of the dataset. We fine-tune the machine translation system on the balanced dataset. We use the WMT18 en-de dataset as processed by Edunov et al. (2018).

**GFST:** Choubey et al. (2021) proposed a method to create a gender-balanced parallel dataset with source- and target-language filtering of pseudo-parallel data. In our work, instead of re-training the model with GFST from scratch, we fine-tune the initially trained model with the target-filtered data. We use the same news2018 corpus used in the original work for the monolingual corpus.

**Handcrafted:** Saunders and Byrne (2020) proposed to use a small, high-quality handcrafted parallel dataset containing a balanced combination of gendered pronouns and occupations to fine-tune the machine translation system. We use the handcrafted en2de dataset provided by the authors in our work, which consists of 388 samples.

### 3.3 Implementation Details

We use three pre-trained multilingual model architectures as our backbone for our experiments: M2M-100 (Fan et al., 2020), SMaLL-100 (Mohammadshahi et al., 2022a), which is a knowledge-distilled model from M2M-100, and NLLB-200 (Costa-jussà et al., 2022). Due to resource limitations, we use a 1.2 billion parameter variant for M2M-100 and a 1.3 billion parameter distilled variant for NLLB-200. We train with a batch size of 8 and learning rate of 4e-6 with 200 warmup steps and inverse square root learning rate decay

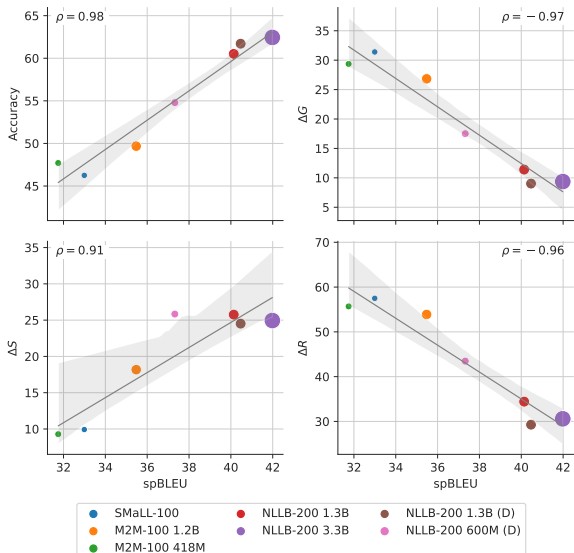

Figure 3: Relationships between translation performance and gender bias metrics of multilingual NMT models. Each point represents the average score of an NMT system on the ten target languages of the WinoMT dataset.

schedule. The hyperparameters for the contrastive learning objective was set to $\lambda = 1.0$ and $\alpha = 0.4$ based on hyperparameter search (refer to Appendix B). We evaluate every 100 training steps and early stop based on the Explicit Accuracy score of the MT-GenEval development set for the fine-tuned language direction. For evaluation, we use beam search of beam size 5 during decoding.

## 4 Results

In this section, we explore the gender bias issue of recent multilingual NMT models and report experimental results of the gender bias mitigation techniques of multilingual machine translation models.

### 4.1 Gender Bias Evaluation of Recent Multilingual NMT Models

We first evaluate existing multilingual NMT models on gender bias and analyze their relationship with their translation performance. We test seven model variants of different sizes based on three model architectures: M2M-100, SMaLL-100, and NLLB-200. For gender bias, we evaluate on the WinoMT dataset for all 10 languages and average the results. Similarly for translation performance, we evaluate on the FLORES-200 devtest set on the same 10 language directions and average the results. For the correlation measure between translation performance and gender bias metrics, we use Pearson's correlation coefficient $\rho$.

| Method | FLORES-200 | | WinoMT | |
| --- | --- | --- | --- | --- |
| | spBLEU$_\uparrow$ | Acc.$_\uparrow$ | $\Delta G_{|\downarrow|}$ | $\Delta S_{|\downarrow|}$ |
| SMaLL-100 | 32.02 | 46.25 | 31.43 | 9.85 |
| M2M-100 | 34.60 | 49.66 | 26.85 | 18.14 |
| NLLB-200 | 40.92 | 62.44 | 9.37 | 24.94 |
| ChatGPT | 27.02$^\dagger$ | 54.25 | 21.09 | 24.95 |

Table 1: Average spBLEU score and WinoMT metrics of NMT systems and ChatGPT (`gpt-3.5-turbo`) on the ten target languages of the WinoMT dataset. $^\dagger$ChatGPT spBLEU score is obtained from Lu et al. (2023).

As shown in Figure 3, we find a strong positive correlation ($\rho = 0.98$) between the translation performance and the gender accuracy. As shown by a negative correlation of $\Delta G$ ($\rho = -0.97$) and $\Delta R$ ($\rho = -0.96$), the accuracy and recall gap between genders are also reduced as translation performance improves. However, the correlation between translation performance and $\Delta S$ is positive ($\rho = 0.91$), implying that better-performing models rely more on occupation stereotypes rather than the original context for gender disambiguation.

Overall, we conclude that recent multilingual models continue to show similar tendencies as MT systems previously reported by Kocmi et al. (2020), with positive correlation with gender accuracy, negative correlation with $\Delta G$, and positive correlation with $\Delta S$. This suggests that the development of NMT systems with a unidimensional focus on performance is insufficient to address the gender bias issue, and active consideration is required.

Recently, ChatGPT[2] has shown remarkable performance in various zero-shot NLP tasks, including machine translation. We evaluate the gender bias of ChatGPT (`gpt-3.5-turbo`) in performing zero-shot machine translation. We use the prompt "Translate the following sentence into *<lang>*. *<sent>*", where *<lang>* is the name of target language and *<sent>* is the source sentence. As shown in Table 1, ChatGPT falls short regarding the spBLEU score but achieves relatively high gender accuracy, surpassing M2M-100 with 54.25. However, we also find that the $\Delta S$ is the highest for ChatGPT, indicating that its translation output often relies on gender stereotypes.

### 4.2 Main Experimental Results

We report the results of fine-tuning multilingual NMT model with our GACL method along with other baseline methods. Specifically, we fine-tune

[2]https://chat.openai.com

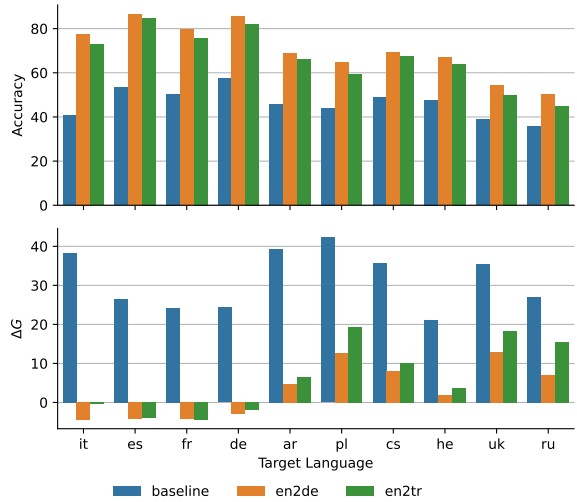

Figure 4: Gender accuracy and $\Delta G$ of individual target languages supported by WinoMT. Results of the SMaLL-100 baseline model, model fine-tuned on en2de dataset, and model fine-tuned on en2tr dataset are reported.

the model on a single language direction and observe its effect on the corresponding language direction (denoted in-domain, ID) as well as other language directions (denoted out-of-domain, OOD). We use the WMT18 en-de dataset (Edunov et al., 2018) for fine-tuning on English to German language direction. For evaluation, all target languages supported by the dataset were evaluated. This involves 10 target languages for WinoMT and 8 for MT-GenEval. Finally, we use the union of the languages covered by the two datasets for evaluating translation performance on the FLORES-200 dataset, which amounts to 12 target languages.

As shown in Table 2, the results on the SMaLL-100 model show that our method, GACL, achieves the greatest improvement in in-domain gender accuracy for both WinoMT and MT-GenEval with 27.3 and 13.0 absolute improvement from the baseline respectively. On the other hand, other baseline methods that were originally proposed for bilingual MT systems proved to be less effective compared to our method. GFST was proved to be the least effective out of the baselines, with less than 3% improvement in gender accuracy, and fine-tuning on the Handcrafted set second was most effective, with 20.6% improvement. Based on these results, we suggest that for multilingual MT models, it is more effective to use a smaller, focused dataset on gender and occupation where the bias of the model is exhibited.

As shown by OOD gender accuracy and $|\Delta G|$

|  | WinoMT | | | | MT-GenEval | | | | FLORES-200 | |
|--|--|--|--|--|--|--|--|--|--|--|
|  | ID | | OOD | | ID | | OOD | | ID | OOD |
| Method | Acc.↑ | $\Delta G_{|\downarrow|}$ | Acc.↑ | $|\Delta G|_{\downarrow}$ | E-Acc.↑ | E-$\Delta G_{|\downarrow|}$ | E-Acc.↑ | $|E\text{-}\Delta G|_{\downarrow}$ | spBLEU↑ | spBLEU↑ |
| Baseline | 57.4 | 24.2 | 45.0 | 32.2 | 54.7 | 10.0 | 42.3 | 27.1 | **36.0** | **32.7** |
| Balanced | 72.9 | 3.8 | 49.7 | 22.4 | 56.7 | 8.3 | 44.1 | 23.9 | 35.4 | 32.4 |
| GFST | 59.9 | 20.9 | 46.2 | 31.1 | 57.3 | 10.3 | 41.9 | 27.2 | 35.5 | 32.4 |
| Handcrafted | 78.0 | **−1.5** | 52.7 | 18.4 | 58.7 | 4.0 | 45.1 | 22.4 | 35.8 | **32.7** |
| *GACL (Ours)* | **84.7** | −3.7 | **65.5** | **8.1** | **67.7** | **−2.0** | **56.2** | **12.9** | **36.0** | **32.7** |

Table 2: Main experimental results on the WinoMT, MT-GenEval, and FLORES-200 datasets for the SMaLL-100 model. The in-domain (ID) setting signifies the en-de language direction in which the model is fine-tuned, while the out-of-domain (OOD) setting encompasses the remaining language directions supported by the dataset.

|  | WinoMT | | | | MT-GenEval | | | | FLORES-200 | |
|--|--|--|--|--|--|--|--|--|--|--|
| Method | Acc.↑ | $|\Delta G|_{\downarrow}$ | $|\Delta S|_{\downarrow}$ | $|\Delta R|_{\downarrow}$ | E-Acc.↑ | $|E\text{-}\Delta G|_{\downarrow}$ | Acc.↑ | $|\Delta G|_{\downarrow}$ | spBLEU↑ | ChrF++↑ |
| **SMaLL-100** | | | | | | | | | | |
| Baseline | 46.2 | 31.4 | 9.9 | 57.5 | 43.8 | 25.0 | 57.9 | 25.6 | **33.0** | **52.3** |
| *GACL (Ours)* | **67.4** | **7.7** | **6.4** | **18.4** | **57.6** | **11.5** | **73.2** | **11.1** | **33.0** | 52.2 |
| - with $\mathcal{L}_{MT}$ Only | 52.0 | 20.6 | 9.3 | 44.2 | 45.7 | 22.0 | 61.3 | 21.6 | 32.6 | 51.8 |
| - with $\mathcal{L}_{GC}$ Only | 63.5 | 9.0 | 7.5 | 21.7 | 55.3 | 13.0 | 71.6 | 12.7 | 32.5 | 51.7 |
| **M2M-100** | | | | | | | | | | |
| Baseline | 49.7 | 26.8 | 18.2 | 53.9 | 44.9 | 23.5 | 58.0 | 23.9 | **35.5** | **53.6** |
| *GACL (Ours)* | **71.4** | **6.4** | **7.3** | **15.5** | **59.3** | **8.8** | **73.3** | **9.2** | 34.9 | 53.1 |
| **NLLB-200** | | | | | | | | | | |
| Baseline | 61.7 | 9.0 | 24.5 | 29.3 | 57.1 | 16.7 | 66.4 | 16.8 | **40.5** | **57.2** |
| *GACL (Ours)* | **78.2** | **3.9** | **6.1** | **6.2** | **69.9** | **5.9** | **80.4** | **5.5** | 40.0 | 56.9 |

Table 3: Full experimental results averaged over all covered languages on the WinoMT, MT-GenEval, and FLORES-200 datasets. Evaluated model architectures are SMaLL-100, M2M-100-1.2B, and NLLB-200-1.3B-distilled.

metrics in Table 2, gender bias mitigation strategies also have a positive effect on the unseen target languages during fine-tuning, regardless of the method used. This implies that gender-related information is agnostic to the target language, and the debiasing effects are transferred to other languages. However, while other baseline methods have a much lower improvement in OOD than ID, our approach is almost as effective in OOD. Fine-tuning on the Handcrafted set improves WinoMT accuracy by 20.6% in ID and 7.7% in OOD, while GACL approach improves by 25.7% and 20.5% respectively.

We report the full results on all evaluated metrics and model architectures in Table 3. We observe that applying GACL improves upon all gender accuracy and bias metrics for all evaluated model architectures. Especially, we find that $|\Delta S|$ metric of NLLB-200, which scored highest out of all methods before fine-tuning, is reduced to 6.1, the lowest out of all methods. On the other hand, we find that the spBLEU and ChrF++ metrics for M2M-100 and NLLB-200 drop by an average of 0.5 points. We suggest that catastrophic forgetting did not occur during fine-tuning due to the model's fast convergence. Still, the fine-tuning was long enough to

significantly improve gender-related translations.

### 4.3 Results for Individual Target Languages

We report the individual results for each target language on the WinoMT dataset in Figure 4. To observe the effect of the target language used during GACL fine-tuning, we evaluate and report on two language directions: English to German (en2de) and English to Turkish (en2tr). In contrast with the German language, which has rich gender morphology, the Turkish language is a gender-neutral language that lacks grammatical gender. We use the same TED2020 corpus (Reimers and Gurevych, 2020) for getting gender-balanced training data to rule out the effect of data domain from our experiments.

Results show that the en2de-finetuned model has higher gender accuracy than the en2tr-finetuned model by an average of 3.6%. However, using en2tr is quite effective on improving gender accuracy and reducing $\Delta G$ on all evaluated target languages. Since the target language of Turkish does not contain gender-related words, results suggest that the gender-related knowledge is accessible from the source encoder representations, and our

| Positive | | Negative | P:N | MT-GenEval | |
| Dropout | In-batch | In-batch | Ratio | Acc. | $|\Delta G|$ |
| --- | --- | --- | --- | --- | --- |
| *Baseline* | | | - | 61.01 | 8.95 |
| ✓ | | ✓ | 1:B | 66.82 | 3.11 |
| | ✓ | ✓ | B-1:B | 66.15 | 3.02 |
| ✓ | ✓ | ✓ | B:B | **66.93** | **2.83** |

Table 4: Ablation of different contrastive pair combinations on the MT-GenEval development set. B denotes one-half of the train batch size.

approach is able to mitigate the bias that lies within it.

## 4.4 Ablation Study

We compare the effects of using different contrastive samples in Table 4. First, we observe that using single dropout representation as the positive sample and in-batch sentence representations with different gender as negative samples achieves substantial performance improvement on both gender accuracy and $\Delta G$. We can also see that using just in-batch samples with the same gender as positive samples can similarly improve the gender accuracy. However, we find that incorporating all available samples within the batch for positive samples achieves the best results.

We also perform an ablation study on training with just machine translation loss $\mathcal{L}_{MT}$, which is equivalent to the Baseline method, and just the gender-aware contrastive loss $\mathcal{L}_{GC}$. The results shown in Table 3 show that training with $\mathcal{L}_{MT}$ on a gender-balanced dataset improves over the baseline by a relatively small amount for all metrics. On the other hand, training with just $\mathcal{L}_{GC}$ loss achieves surprisingly high performance on both gender evaluation benchmarks. We point out that training on $\mathcal{L}_{GC}$ only updates the encoder parameters of an encoder-decoder model, and thus having gender-aware contextual embedding in the encoder representations can be effective. We also observed during the fine-tuning process with only $\mathcal{L}_{GC}$ that the performance converges quickly within 200 steps, and upon further fine-tuning, both translation performance and gender accuracy deteriorate very quickly. We thus conclude that losses like $\mathcal{L}_{MT}$ and $\mathcal{L}_{KD}$ are required to prevent catastrophic forgetting during fine-tuning and enable stable training to get optimal convergence on gender-aware contrastive learning.

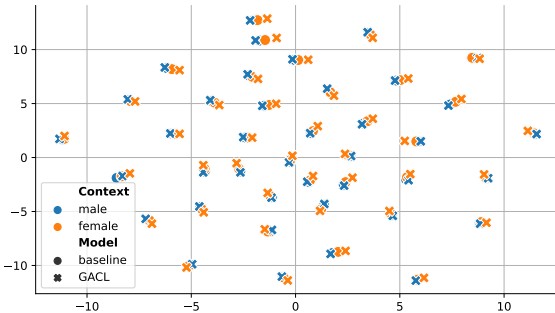

Figure 5: t-SNE visualization of encoder output representation of occupation words. Circle and cross markers denote embeddings of the original SMaLL-100 model and GACL-finetuned variant, respectively. The colors of the marker denote the context of the occupation word.

## 5 Analysis

In this section, we investigate how our GACL fine-tuning method affects the model representations regarding gender. Specifically, we use 40 stereotypical occupation words from the WinoMT dataset, where 20 are stereotypically assumed male while the remaining 20 are assumed female. We label the "stereotypical gender" of an occupation word as defined by this gender assumption. We then construct a sentence with the template "He is *<occupation>*." and "She is *<occupation>*." as encoder input. Here, the pronouns decide the "contextual gender" of the occupation word. Finally, a single contextual representation vector is computed by taking the average of encoder output representations of the tokens that make up the occupation word. We use this same representation extraction process for both the baseline SMaLL-100 model and the SMaLL-100 model fine-tuned with GACL.

To examine the representations, we employ the t-SNE dimension reduction technique (van der Maaten and Hinton, 2008) to visualize the occupation representations in 2 dimensions, as shown in Figure 5. We observe that the representations for each occupation are clustered closely together regardless of the sentence context and model. This shows that the contextual gender has a relatively little contribution to the representation compared to the semantic meaning of the occupation. Also, our fine-tuning method induces a relatively small change, preserving the semantic distinction between the occupations. Finally, we note that the average distance between representations of different contexts is farther apart for the GACL representations (0.59) than the baseline representations (0.19), suggesting that the contrastive objective has

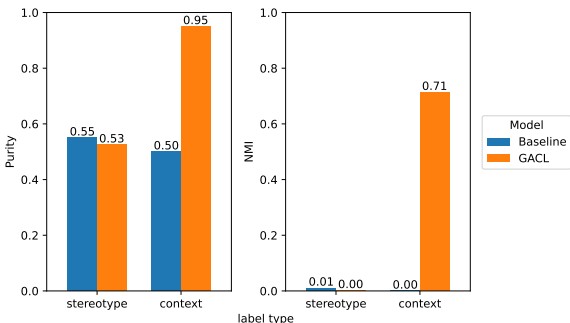

Figure 6: Clustering results on occupation word embeddings based on stereotypical and contextual gender label assignments. Lower scores are better for stereotypical gender labels, and higher scores are better for contextual gender labels.

guided the model to differentiate the occupation based on the gender context.

To investigate how much gender information is encoded within the embeddings in-depth, we perform $k$-means clustering with $k = 2$ on the occupation embeddings, and evaluate the cluster quality based on stereotypical and contextual gender as label assignments. Based on the Purity and Normalized Mutual Information (NMI) metrics, we see that clusters for both models have negligible alignment with the stereotypical gender assignment (Figure 6). On the other hand, we find that clustering based on GACL embeddings is very well aligned with the contextual gender, while the baseline model continues to be misaligned. This shows that GACL representations capture contextual gender information significantly better than the baseline representations.

## 6 Related Works

**Gender Bias in NLP** Chung et al. (2022) point out that the existing English-centric Large Language Models (LLMs) suffer from gender-stereotypical words. However, fixing biases that are deeply ingrained in hidden representations is a challenging task (Gonen and Goldberg, 2019; Orgad and Belinkov, 2022). Previous researchers such as Ravfogel et al. (2020); Kumar et al. (2020); Gaci et al. (2022b) debias the hidden representation using learning-based methods. However, Kumar et al. (2022) point out the limitations of such studies and recommend the use of data augmentation techniques (Webster et al., 2020; Sharma et al., 2021; Lauscher et al., 2021). In addition to aforementioned research, Attanasio et al. (2022) and Gaci et al. (2022a) focus on biased attention weights, and

Cheng et al. (2021) and He et al. (2022) use a contrastive learning scheme to reduce bias in sentence embeddings and language models respectively.

**Multilingual Machine Translation** Studies such as mBERT (Pires et al., 2019) and XLM-R (Goyal et al., 2021) have shown that it is possible to train language models on multiple languages simultaneously, a method referred to as multilingual training. Recent research has proven that multilingual training contributes to a positive impact on NMT (Aharoni et al., 2019; Tran et al., 2021; Chiang et al., 2022). According to Carrión-Ponz and Casacuberta (2022), by training multilingual NMT models further with a few-shot regularization, a decrease in the performance can be prevented. Knowledge distillation also helps NMT models preserve their original performance (Shao and Feng, 2022).

**Gender Bias in Machine Translation** Bilingual NMT models have been shown to be easily exposed to gender biases (Prates et al., 2020). Correspondingly, Zhao et al. (2018); Choubey et al. (2021); Currey et al. (2022) employ data augmentation-based approach to reduce gender bias of MT models. In addition, Saunders and Byrne (2020) propose utilizing a transfer-learning method, and Savoldi et al. (2021) develop a unified framework to tackle the biases. However, these works mostly do not consider multilingual NMT models that support multiple language directions.

Alternatively, Fleisig and Fellbaum (2022) propose an adversarial learning framework to mitigate gender bias in machine translation models by removing gender information when the input has masked gender context. Our approach, on the other hand, injects the correct contextual gender information from encoder output contrastively when given inputs have gender contexts.

In the case of multilingual machine translation, Costa-jussà et al. (2022) have shown that the shared encoder-decoder architecture of multilingual NMT models has a negative effect on the gender bias. Also, Mohammadshahi et al. (2022b) investigate how the multilingual model compression affects gender bias. Contemporary work by Cabrera and Niehues (2023) examines gender preservation in zero-shot multilingual machine translation. However, no existing work that the authors are aware of specifically considers mitigating the unambiguous gender bias of multilingual NMT models for multiple language directions simultaneously.

# 7 Conclusion

In this work, we conducted an investigation into the gender bias of multilingual NMT models, specifically focusing on unambiguous cases and evaluating multiple target languages. Our findings indicated that even state-of-the-art multilingual NMT systems tend to exhibit a preference for gender stereotypes in translations. We then proposed a novel debiasing method, Gender-Aware Contrastive Learning (GACL), which injects contextually consistent gender information into latent embeddings. Our experiments demonstrated that GACL effectively improves gender accuracy and reduces gender performance gaps in multilingual NMT models, with positive effects extending to target languages not included in fine-tuning. These findings highlight the importance of addressing gender bias in machine translation and provide a promising approach to mitigate it in multilingual NMT systems.

## Limitations

The gender debiasing process in this study relies on a curated list of gender word pairs to identify and filter gendered terms in the dataset. However, this approach may not cover the whole of the gendered terms present in the world. The limited coverage of gendered terms could potentially introduce biases and inaccuracies in the evaluation results, as certain gendered terms may be missed or not appropriately accounted for.

Furthermore, our method only deals with binary gender, and do not consider the possible representations of non-binary genders and their bias in translation. As languages vary in their use of grammatical gender and they often lack clearly defined rules or established linguistic structures for non-binary genders, it is especially challenging to evaluate and mitigate bias in this context. This limitation highlights the need for further research and consideration of more diverse gender representations in bias mitigation.

While we extend gender bias evaluation to multilingual settings, this study is still limited to the provided target languages, which predominantly include medium-to-high resource languages. Due to a lack of evaluation data, the evaluated source language is also limited to English. Consequently, the findings and conclusions may not be representative of the gender biases present in languages that are not included in the evaluation. The limitations in language coverage may restrict the generalizability of the study's results to a broader linguistic context.

The focus of this study is primarily on evaluating gender bias in unambiguous settings where the intended gendered terms are clear. However, the investigation of gender bias in ambiguous settings, where the gendered term can have multiple interpretations, is not addressed in this study. Consequently, the study does not provide insights into potential biases in ambiguous gendered language use, which can also contribute to societal biases and stereotypes.

Our work focuses on recent NMT models based on the encoder-decoder architecture, and hence the effect of our method on decoder-only NMT models remains unverified. Nevertheless, our approach is applicable to other model architectures as long as we can aggregate a representation of the source input sentence. Specifically for decoder-only architectures, one feasible strategy would be to pool the decoder model outputs of tokens up to given source input sentence. As decoder-only large language models such as ChatGPT are increasingly being considered for the MT task, we believe this is an interesting direction for future work.

## Ethical Considerations

Our work attempts to reduce bias and misrepresentations in translation of masculine and feminine gendered referents. Our methodology and evaluation has been limited to considering binary genders, which overlooks non-binary genders and correspondingly doesn't consider bias in gender-inclusive and gender-neutral translations. Possible mitigations include extending the translation data to incorporate sentences with gender-neutral inflections and defining a separate gender pseudo-label for applying proposed contrastive loss. The lack of flexibility in gender could also be mitigated by extending our work to controlled generation where the preferred gender inflection is given as input.

Furthermore, in our work, the evaluated source language was limited to English, and evaluated target languages are mostly of high-resource. Correspondingly, our work may under-represent bias found in unevaluated low-resource languages. However, the findings in our work show potential gender debiasing effects transferring to non fine-tuned languages, and extending gender bias evaluation resources to include low-resource languages may help analyze and mitigate gender bias

for those languages.

## Acknowledgements

This research was supported by the MSIT(Ministry of Science, ICT), Korea, under the High-Potential Individuals Global Training Program)(2022-00155958) supervised by the IITP(Institute for Information & Communications Technology Planning & Evaluation) (Contribution: 50%), and Institute of Information & communications Technology Planning & Evaluation(IITP) grant funded by the Korea government(MSIT) [No. 2022-0-00184, Development and Study of AI Technologies to Inexpensively Conform to Evolving Policy on Ethics], and Institute of Information & communications Technology Planning & Evaluation (IITP) grant funded by the Korea government(MSIT) [NO.2021-0-01343, Artificial Intelligence Graduate School Program (Seoul National University)], and Microsoft Research Asia. K. Jung is with Automation and Systems Research Institute (ASRI), Seoul National University.

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

## A  Experimental Details

### A.1  ChatGPT Generation

For our experiments using ChatGPT, we use the API provided by OpenAI for generating text, using the model gpt-3.5-turbo. For a small number of cases, ChatGPT generated multiple lines of text delimited by newline characters, leading to errors in source-target alignment during WinoMT evalution. For these cases, we split the sentence based on the newline character and take the first sentence as the translation.

### A.2  Data Preprocessing Details

In our work, we use the cleaned WMT18 en-de dataset, as pre-processed by Edunov et al. (2018). By filtering for sentences with gender-related words, we find 415,401 masculine and 86,431 feminine sentence sets. Next, masculine sentence sets are undersampled with random sampling while all samples from feminine set are used to create a final balanced set of total 2*86,431=172,862 samples. We do not perform additional processing to handle other domain differences except gender.

### A.3  Fine-tuning Details

For our experiments, hyperparameter search was done manually with learning rate from {2e-6, 4e-6, 8e-6}, batch size from {4, 8, 16, 32}, and learning rate warmup steps from {100, 200, 400} based on fine-tuning GACL on the SMaLL-100 architecture and metric based on explicit accuracy on the MT-GenEval development set. Our final selection of hyperparameters is then used for all experiments in our paper. One exception for fine-tuning GACL with NLLB-200 architecture is we set the learning rate to 8e-6 due to its slower convergence during training in comparison to other architectures. For fine-tuning with our GACL method, we use balanced random sampling so that the number of sentences for each gender are equal within one mini-batch.

The fine-tuned dataset size and number of fine-tuning steps before early stopping is shown in Table 5. We notice that Balanced and GFST data augmentation based methods trained for more than one thousands steps before early stopping. On the other hand, our GACL method and Handcrafted method stopped fine-tuning within one thousand steps.

We fine-tuned the SMaLL-100 model on 1 NVIDIA A6000 GPU, and fine-tuned M2M-100 1.2B and NLLB-200 1.3B distilled models on 1

| Method | Dataset size | # Steps |
|---|---|---|
| **SMaLL-100** | | |
| Balanced | 172,862 | 3,700 |
| GFST | 1,802,832 | 1,900 |
| Handcrafted | 388 | 200 |
| GACL | 172,862 | 700 |
| - with $\mathcal{L}_{GC}$ only | - | 200 |
| **M2M-100** | | |
| GACL | 172,862 | 400 |
| **NLLB-200** | | |
| GACL | 172,862 | 800 |

Table 5: Dataset size and number of training steps before early stopping for the fine-tuning experiments in our work.

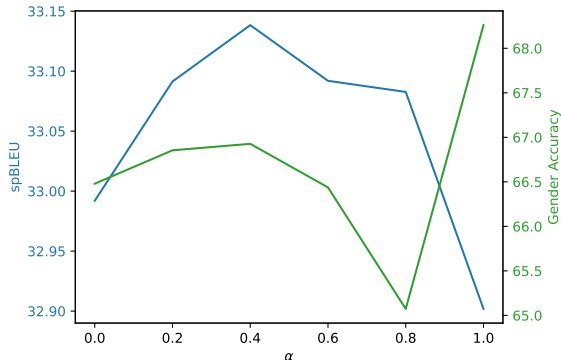

Figure 7: Experimental results on tuning hyperparameter $\alpha$. the spBLEU scores are the average score on 10 language directions on FLORES200 development set and gender accuracy is the average explicit accuracy on the MT-GenEval development set.

NVIDIA A100 80GB GPU. We use the pre-trained model checkpoints downloaded from the Huggingface website.[3]

## B  Additional Experiments

### B.1  Many2many Translation Performance

The 12 target languages covered by the WinoMT and MT-GenEval are mostly categorized as high-resource languages (Mohammadshahi et al., 2022a). Thus, we extend our FLORES-200 evaluation to languages of low and medium resources for both source and target languages to more accurately analyze the impact of our approach on multilingual translation performance.

For this experiment, we use the four resource levels defined by Mohammadshahi et al. (2022a): High (H), Medium (M), Low (L) and Very Low

---

[3] https://huggingface.co/

| | FLORES-200 spBLEU | | | | | | | | | | | | | | | | |
|---|---|---|---|---|---|---|---|---|---|---|---|---|---|---|---|---|---|
| Method | VL2VL | VL2L | VL2M | VL2H | L2VL | L2L | L2M | L2H | M2VL | M2L | M2M | M2H | H2VL | H2L | H2M | H2H | X2X |
| **SMALL-100** | | | | | | | | | | | | | | | | | |
| Baseline | 26.6 | 14.5 | 21.8 | 15.3 | 17.7 | 9.4 | 13.7 | 11.3 | 22.6 | 12.2 | 19.7 | 11.7 | 21.2 | 12.6 | 15.3 | 12.2 | 16.1 |
| GACL | 26.8 | 13.9 | 21.7 | 14.8 | 17.1 | 8.7 | 13.5 | 10.8 | 22.6 | 11.7 | 19.8 | 11.3 | 21.1 | 12.1 | 15.2 | 11.6 | 15.8 |
| - w/o $\mathcal{L}_{KD}$ | 26.7 | 13.1 | 21.4 | 14.0 | 16.5 | 7.7 | 12.9 | 9.9 | 22.4 | 10.9 | 19.7 | 10.7 | 20.8 | 11.3 | 14.8 | 10.6 | 15.2 |
| **M2M-100** | | | | | | | | | | | | | | | | | |
| Baseline | 30.0 | 12.7 | 22.9 | 13.1 | 19.7 | 8.3 | 14.0 | 11.4 | 25.5 | 10.7 | 20.2 | 11.0 | 22.4 | 10.7 | 16.3 | 9.9 | 16.2 |
| GACL | 29.8 | 12.0 | 22.3 | 12.8 | 19.7 | 7.8 | 13.7 | 11.1 | 25.4 | 10.1 | 19.8 | 10.6 | 22.2 | 10.0 | 15.7 | 9.5 | 15.8 |
| **NLLB-200** | | | | | | | | | | | | | | | | | |
| Baseline | 31.9 | 26.0 | 26.6 | 25.8 | 30.2 | 25.5 | 25.9 | 25.2 | 28.7 | 24.0 | 24.2 | 23.4 | 30.3 | 25.7 | 25.6 | 24.0 | 26.4 |
| GACL | 31.4 | 25.7 | 25.9 | 25.1 | 29.8 | 25.1 | 25.2 | 24.6 | 28.3 | 23.5 | 23.5 | 22.7 | 29.8 | 25.2 | 24.7 | 23.2 | 25.9 |

Table 6: Many2many translation performance evaluation by resource level on FLORES-200 using the spBLEU metric. X2X represents the total average score.

| | FLORES-200 ChrF++ | | | | | | | | | | | | | | | | |
|---|---|---|---|---|---|---|---|---|---|---|---|---|---|---|---|---|---|
| Method | VL2VL | VL2L | VL2M | VL2H | L2VL | L2L | L2M | L2H | M2VL | M2L | M2M | M2H | H2VL | H2L | H2M | H2H | X2X |
| **SMALL-100** | | | | | | | | | | | | | | | | | |
| Baseline | 48.0 | 34.4 | 35.3 | 35.9 | 38.7 | 27.7 | 27.4 | 30.2 | 44.7 | 31.9 | 33.7 | 31.5 | 43.1 | 32.4 | 28.7 | 33.0 | 34.8 |
| GACL | 48.1 | 33.5 | 35.1 | 35.2 | 38.0 | 26.6 | 26.9 | 29.2 | 44.5 | 30.9 | 33.6 | 30.8 | 43.0 | 31.5 | 28.5 | 32.0 | 34.2 |
| - w/o $\mathcal{L}_{KD}$ | 48.0 | 32.1 | 34.7 | 34.1 | 37.1 | 24.7 | 25.8 | 27.6 | 44.3 | 29.5 | 33.4 | 29.8 | 42.6 | 30.0 | 27.8 | 30.6 | 33.3 |
| **M2M-100** | | | | | | | | | | | | | | | | | |
| Baseline | 50.4 | 29.9 | 35.7 | 31.7 | 39.0 | 24.0 | 26.0 | 29.8 | 46.7 | 27.9 | 33.5 | 29.4 | 43.2 | 27.7 | 29.3 | 28.8 | 33.3 |
| GACL | 50.3 | 29.1 | 35.2 | 31.5 | 39.1 | 23.2 | 25.7 | 29.3 | 46.7 | 27.1 | 33.1 | 29.1 | 43.0 | 26.8 | 28.8 | 28.2 | 32.9 |
| **NLLB-200** | | | | | | | | | | | | | | | | | |
| Baseline | 51.6 | 45.3 | 38.5 | 45.9 | 50.0 | 44.6 | 37.9 | 45.3 | 49.0 | 43.7 | 36.6 | 43.8 | 50.4 | 45.2 | 37.9 | 44.6 | 44.4 |
| GACL | 51.2 | 45.1 | 38.1 | 45.4 | 49.7 | 44.4 | 37.4 | 44.7 | 48.7 | 43.2 | 36.0 | 43.2 | 50.1 | 44.8 | 37.3 | 44.0 | 44.0 |

Table 7: Many2many translation performance evaluation by resource level on FLORES-200 using the ChrF++ metric. X2X represents the total average score.

(VL), and evaluate many2many translation performance using spBLEU. Due to resource limitations, we sample 5 languages for each resource group and evaluate across all sampled language directions. The languages used are as follows: High resource languages include French, German, Italian, Russian, and Spanish (Latin America). Medium resource languages are Arabic, Bulgarian, Chinese (Simplified), Korean, and Turkish. Low resource languages are Afrikaans, Amharic, Bosnian, Cebuano, and Kazakh. Very Low resource languages are Belarusian, Croatian, Filipino (Tagalog), Nepali, and Occitan. The averaged results by resource group are reported in Tables 6 and 7.

## B.2 Ablation Results on Hyperparameter $\alpha$

We report the effects of changing the hyperparameter $\alpha$ used to determine the relative weight between $\mathcal{L}_{MT}$ and $\mathcal{L}_{KD}$ in the joint training loss we employed during fine-tuning.

As shown in Figure 7, we find that for translation performance, setting $\alpha$ to 0.4 performs the best, while using a single loss of either $\mathcal{L}_{MT}$ (i.e. $\alpha = 0$) and $\mathcal{L}_{KD}$ (i.e. $\alpha = 1$) performs slightly worse. For gender accuracy, we find that trends are not very clear, with $\alpha$ set to 1.0 being the most effective and $\alpha$ of 0.4 second most effective. Based on these findings, we choose to use $\alpha$ value of 0.4 for the rest of the experiments in this paper.

## B.3 Relationship between translation performance and gender bias metrics for each language

In Figure 8, we report results on the relationship between translation performance and gender bias metrics for each language. We observe similar correlations between translation performance and gender bias metrics of multilingual MT systems across each independent target languages. However, the slope of the correlation differs by the target language.

## B.4 Gender bias evaluation results for each target language

We report the evaluation results on WinoMT and MT-GenEval datasets for each of the supported target languages individually in Tables 8 and 9. For all evaluations, the source language is fixed to English, as it is the only provided source language for the datasets.

## B.5 Statistical significance tests

We share the results on statistical significance testing between our GACL model (Table 3; row 2) and our ablation fine-tuned with $\mathcal{L}_{GC}$ only (Table 3; row 4) which scored closely in accuracy scores. We conduct a paired randomized permutation test with the number of resamples $N$ set to 100,000. The p-values from the test are shown in Table 10.

| Target lang. | de | | ru | | fr | | it | | es | | uk | | he | | ar | | pl | | cs | | AVG | |
|---|---|---|---|---|---|---|---|---|---|---|---|---|---|---|---|---|---|---|---|---|---|---|
| | Acc. | $\Delta G$ | Acc. | $\Delta G$ | Acc. | $\Delta G$ | Acc. | $\Delta G$ | Acc. | $\Delta G$ | Acc. | $\Delta G$ | Acc. | $\Delta G$ | Acc. | $\Delta G$ | Acc. | $\Delta G$ | Acc. | $\Delta G$ | Acc. | $\Delta G$ |
| Baseline | 57.4 | 24.2 | 35.8 | 27.0 | 50.3 | 24.3 | 40.7 | 38.2 | 53.3 | 26.4 | 38.8 | 35.5 | 47.7 | 21.0 | 45.6 | 39.5 | 44.2 | 42.3 | 48.9 | 35.5 | 46.2 | 31.4 |
| Balanced | 72.9 | 3.8 | 38.8 | 22.6 | 59.6 | 7.9 | 43.4 | 29.1 | 64.7 | 8.0 | 40.1 | 32.5 | 49.2 | 17.0 | 50.3 | 28.1 | 46.8 | 34.0 | 54.3 | 22.8 | 52.0 | 20.6 |
| GFST | 59.9 | 20.9 | 36.3 | 27.8 | 54.2 | 17.0 | 41.4 | 38.3 | 55.9 | 21.8 | 39.6 | 36.2 | 47.9 | 22.0 | 46.5 | 40.9 | 43.8 | 41.8 | 49.8 | 34.5 | 47.5 | 30.1 |
| Handcrafted | 78.0 | −1.5 | 38.1 | 22.7 | 63.9 | 3.2 | 49.4 | 19.3 | 69.8 | 2.6 | 42.3 | 28.9 | 51.5 | 13.3 | 51.6 | 25.6 | 50.6 | 29.8 | 57.1 | 20.1 | 55.2 | 16.4 |
| GACL (Ours) | 84.8 | −3.7 | 46.8 | 10.8 | 78.4 | −7.8 | 67.4 | −2.5 | 85.5 | −5.1 | 52.5 | 15.0 | 64.3 | 2.8 | 66.1 | 4.7 | 61.1 | 14.4 | 67.4 | 9.7 | 67.4 | 3.9 |

Table 8: Accuracy and $\Delta G$ scores on the 10 target languages of the WinoMT dataset.

| Target lang. | de | | ru | | fr | | it | | es | | pt | | ar | | hi | | AVG | |
|---|---|---|---|---|---|---|---|---|---|---|---|---|---|---|---|---|---|---|
| | Acc. | $\Delta G$ | Acc. | $\Delta G$ | Acc. | $\Delta G$ | Acc. | $\Delta G$ | Acc. | $\Delta G$ | Acc. | $\Delta G$ | Acc. | $\Delta G$ | Acc. | $\Delta G$ | Acc. | $\Delta G$ |
| Baseline | 62.3 | 10.0 | 68.7 | 20.0 | 55.3 | 23.0 | 49.0 | 39.0 | 56.3 | 27.0 | 53.0 | 27.7 | 65.7 | 14.3 | 52.7 | 38.7 | 57.9 | 25.0 |
| Balanced | 65.7 | 8.3 | 68.7 | 19.0 | 57.3 | 21.3 | 57.0 | 34.3 | 60.7 | 19.3 | 56.3 | 25.0 | 72.0 | 10.7 | 52.7 | 37.7 | 61.3 | 22.0 |
| GFST | 65.0 | 10.3 | 67.3 | 22.0 | 53.3 | 23.0 | 48.7 | 38.3 | 55.3 | 26.0 | 53.0 | 29.0 | 67.0 | 14.3 | 50.7 | 37.7 | 57.5 | 25.1 |
| Handcrafted | 66.7 | 4.0 | 69.7 | 17.0 | 57.0 | 18.7 | 56.0 | 33.0 | 61.0 | 19.7 | 57.7 | 22.3 | 70.7 | 12.0 | 53.7 | 34.3 | 61.5 | 20.1 |
| GACL (Ours) | 76.0 | −2.0 | 84.7 | 5.3 | 65.3 | 9.0 | 71.7 | 18.0 | 72.7 | 9.0 | 70.7 | 10.0 | 85.3 | 7.7 | 59.0 | 31.3 | 73.2 | 11.0 |

Table 9: Accuracy and $\Delta G$ scores on the 8 target languages of the MT-GenEval test set.

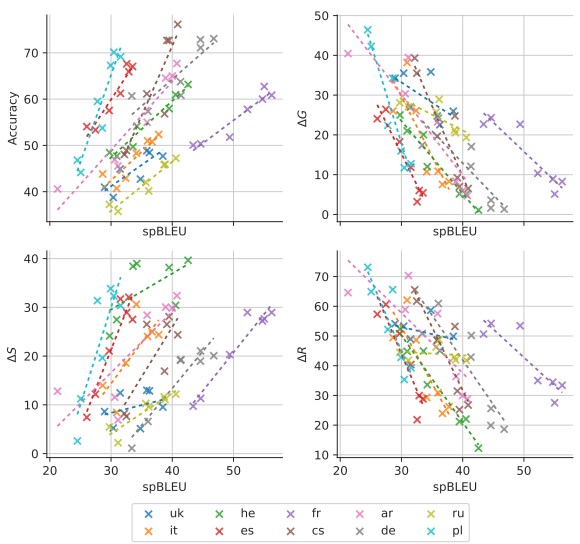

Figure 8: Relationships between translation performance and gender bias metrics of multilingual NMT model for various evaluated target languages.

We found that the GACL method shows higher accuracy than the ablation on 10 out of 10 evaluated languages on WinoMT dataset and 4 out of 8 evaluated languages on MT-GenEval dataset directions with statistical significance of p-value less than 0.05.

## B.6 Gender evaluation on the secondary entity of the WinoMT dataset

The WinoMT dataset is comprised of sentences that mention two entities; one entity has an unambiguous gender indicated by a coreferential pronoun, while the gender of the other entity is ambiguous. In this subsection, we evaluate the effect of gender debiasing methods on this secondary, gender-unspecified entity in the WinoMT dataset. We use the variation dataset proposed by Saunders et al.

| Dataset | Target Lang. | P-value |
|---|---|---|
| WinoMT | de | $< 1e-5$ |
| | ru | $6e-5$ |
| | fr | $< 1e-5$ |
| | it | $< 1e-5$ |
| | es | $< 1e-5$ |
| | uk | $< 1e-5$ |
| | he | $< 1e-5$ |
| | ar | $< 1e-5$ |
| | pl | $< 1e-5$ |
| | cs | $< 1e-5$ |
| MT-GenEval | ru | $1e-2$ |
| | es | $4e-2$ |
| | fr | $1e-1*$ |
| | it | $1e-2$ |
| | ar | $2e-1*$ |
| | de | $4e-1*$ |
| | hi | N/A |
| | pt | $4e-5$ |

Table 10: P-values from randomized pairwise permutation test between accuracy scores of GACL and ablation. * denotes evaluations that are not statistically significant with respect to threshold of 0.05.

| Method | $\text{Acc}_{prim.}$ | $\text{Acc}_{sec.}$ | $\Delta$ |
|---|---|---|---|
| Baseline | 46.3 | 48.6 | 2.3 |
| GFST | 47.5 | 52.2 | 4.7 |
| Handcrafted | 55.2 | 59.0 | 3.8 |
| GACL (Ours) | 67.4 | 72.7 | 5.3 |

Table 11: Gender "Accuracy" of the primary and secondary entities in the WinoMT dataset.

(2020) and show the results in Table 11. We found that all previous methods as well as ours lead to an consistent increase in gender accuracy of secondary entity by about 5% compared to gender accuracy of primary entity, which is consistent with findings by Saunders et al. (2020). Note that none of the evaluated methods, including ours, explicitly account for entities with ambiguous gender and this issue is left for future research.