# OpenReview forum: "Target-Agnostic Gender-Aware Contrastive Learning for Mitigating Bias in Multilingual Machine Translation"
_EMNLP/2023/Conference — EMNLP 2023 Main_

### Official Review · Reviewer_KfGt · 2023-07-25

**Soundness:** 4

**Excitement:**

4: Strong: This paper deepens the understanding of some phenomenon or lowers the barriers to an existing research direction.

**Paper Topic And Main Contributions:**

from the abstract:  the paper specifically targets gender bias for multilingual systems and studies cases where there is a single correct (i.e. unambiguous) translation.   A straightforward modelling technique termed Gender-Aware Contrastive Learning is proposed that encodes contextual gender information.    This is a source-side technique,  in that it influences the encoder, and so offers improved gender translation even into target languages not seen in training (fine tuning).    The Gender-Aware Contrastive Learning technique is described in section 2.2 , and is motivated by recent in-batch contrasting training techniques.  Regularisation in fine tuning is also needed,  as is confirmed by ablation experiments in section 4.4.   Improvements in gender accuracy across a range of target languages and architectures are reported using the usual suite of metrics, including a newly proposed Explicit Accuracy.

**Questions For The Authors:**

A.  Line 135 states:  `Given h_i as the encoder embedding of the source sentence, we define positive samples to be the set of representations that have the same gender as h_i'.   Can this be clarified?   If h_i is a sentence embedding,  its not clear how it can have a gender.    Some explanation as to how gender is extended to the sentence level,  or perhaps a clarification that h_i is a word-level representation (?), would make this more clear.   Given that section 2.2 contains the main novelty in the paper, it should be made more explicit.

B. The presentation seems to emphasize that the technique is suitable for encoder-decoder models, and modifying the encoder embedding makes for a nice presentation.  But couldn't the technique also be applied in a decoder architecture?

C. More a comment than a question, but readers might be as interested in BLEURT as spBLEU and ChrF++ .

**Reasons To Accept:**

The paper presents a fairly simple technique that gives good improvements to accuracy in translation for multilingual NMT models.   The technique has a practical flavour: the focus on the source embedding, rather than the target language,  yields improvements in gender accuracy into multiple target languages,  with good generalisation from the language pairs used in training with the contrastive loss.   A good range of language pairs and multilingual models are evaluated.   The paper also has many interesting observations, e.g. that ChatGPT often relies on gender stereotypes in its translation output (line 343).   The chosen baseline techniques are suitable for the proposed technique.

**Reasons To Reject:**

As noted in the limitations section,  the evaluated source language is limited to English.   I can't recommend an easy fix for this, given what evaluation resources are available,  but it is a significant limitation.   Another limitation is the focus on encoder-decoder architectures (line 114).   Many recent LLMs will not be able to make use of this method.

**Reproducibility:**

3: Could reproduce the results with some difficulty. The settings of parameters are underspecified or subjectively determined; the training/evaluation data are not widely available.

**Reviewer Confidence:**

3: Pretty sure, but there's a chance I missed something. Although I have a good feel for this area in general, I did not carefully check the paper's details, e.g., the math, experimental design, or novelty.

**Typos Grammar Style And Presentation Improvements:**

An example of the 'Explicit Accuracy' (line 226)  showing how it differs from normal accuracy (line 214) would make it easier to follow.

Figure 3 shows average scores of a NMT system on ten target languages of WinoMT.    Is the regression done against the individual scores of individual systems on specific LPs,  or is it against the average scores?    Its difficult to interpret this figure.

Note that Table 2 column headings in the 'Method' row have misplaced | | 's for the \Delta G 's

---

> ### Author Rebuttal · Authors · 2023-08-29
>
> Thank you for highlighting many strengths of our paper and pointing out parts that are unclear. We respond to your comments below.
>
> > As noted in the limitations section, the evaluated source language is limited to English.
>
> We focused our work on extending evaluation to multiple target languages in a multilingual setting first and evaluated on a total of 12 target languages, which is a great increase in coverage in comparison to previous MT debiasing literature that usually considered bilingual models for three to five target languages (Choubey et al 2021, Saunders and Bryne 2020). However, our work is definitely extensible to source languages other than English, such as Chinese, if there are means to detect gendered terms in the sentence, as the proposed contrastive loss itself is not dependent on any specific language.
>
> >  Another limitation is the focus on encoder-decoder architectures
>
> Our work focuses on encoder-decoder architectures as most of the best-performing multilingual NMT models employed an encoder-decoder architecture in our review of literature. However, our approach is applicable to other model architectures as long as we can aggregate a representation of the source language input. Specifically for decoder-only architectures, an option would be to pool the decoder model outputs of tokens up to given source language input. As LLMs are increasingly being considered for the MT task, we believe this is an interesting direction for future work.
>
> > Q A. Line 135 states: `Given h_i as the encoder embedding of the source sentence, we define positive samples to be the set of representations that have the same gender as h_i'. Can this be clarified?
>
> h_i is a sentence embedding, and we use the gendered word mentioned within the source sentence as the gender pseudo-label for the sentence embedding during contrastive learning. While it might seem unintuitive to assign a gender to a sentence, we use sentence-level instead of word-level representations because it is difficult to know in advance which words in the given sentence will be translated to have a gender-specific marking and which will not, and this also varies by the target language. Therefore, we rely on indirectly incorporating gender information through the sentence embedding instead, which is computed by pooling all word representations. We will update section 2.2 to make this more explicit.
>
> > Q B. ... couldn't the technique also be applied in a decoder architecture?
>
> Yes, we believe our method is applicable to decoder-only models by computing representations based on decoder output of the given source sentence input. However, we focused our work on studying encoder-decoder architecture as most state-of-the-art multilingual NMT models were based on them.
>
> > Q C. ... readers might be as interested in BLEURT as spBLEU and ChrF++ .
>
> Thank you for the recommendation. We evaluated on spBLEU and ChrF++ primarily to allow comparison with published results from previous works on multilingual NMT. While extending evaluation with BLEURT would strengthen the comparison, due to the large number of language directions we were evaluating on and limited resources, we decided to focus more on the evaluation metrics related to gender bias.
>
> > Figure 3 shows average scores of a NMT system on ten target languages of WinoMT. Is the regression done against the individual scores of individual systems on specific LPs, or is it against the average scores?
>
> The regression is done against the average scores for Figure 3. We will update our paper to make this clear. We also have regression plots for individual scores of individual systems for each target language in Figure 8 of Appendix.
>
> Overall, thank you for your comments and hope our responses have sufficiently addressed your concerns.

---

### Official Review · Reviewer_3Bkt · 2023-07-25

**Soundness:** 4

**Ethical Concerns:**

Yes

**Excitement:**

4: Strong: This paper deepens the understanding of some phenomenon or lowers the barriers to an existing research direction.

**Justification For Ethical Concerns:**

No impact statement is provided, although the paper deals with a dimension (gender) that directly impacts users.

**Missing References:**

Lines 222-233 define gender accuracy exactly as defined in:
M. Gaido et al. 2020. "Breeding Gender-aware Direct Speech Translation Systems", Proc. of COLING 2020.


Contemporaneous work on gender bias in multilingual MT, which might be included in a camera ready:
L Cabrera, J. Niehues. 2023. "Gender Lost In Translation: How Bridging The Gap Between Languages Affects Gender Bias in Zero-Shot Multilingual Translation", In Workshop on Gender-Inclusive Translation Technologies (GITT) at EAMT 2023.


**Paper Topic And Main Contributions:**

The paper introduces a novel strategy to mitigate gender bias in multilingual machine translation models by exploiting contrastive learning. The proposed approach consists in adding a new loss that takes the encoder outputs for all the samples in the batch and forces a higher similarity for those tensors that contain a referent of the same gender. In addition to this loss, the finetuning process accounts for other two losses: the traditional cross entropy and a knowledge distillation loss, which aim at avoiding the catastrophic forgetting problem. The results on different benchmarks demonstrate the ability of the approach in reducing gender bias, while retaining translation quality.

**Questions For The Authors:**

In lines 127-131 it is stated that the training data is created so that it is balanced. Is this a requirement for the method? Has it been tested with unbalanced data and resampling?
Was the method tested in the context of ambiguous referents?

**Reasons To Accept:**

 1. The approach demonstrates effective, compares favourably to other strategies, and is efficient.
 2. The evaluation is convincing and complete, with detailed analyses explaining the strengths of the proposed approach.

**Reasons To Reject:**

 1. An impact statement is missing, while it would probably be required to delineate the effect of the techniques on NMT users, and how it relates to groups that are neglected in the study (e.g., non-binary individuals).
 2. It would be interesting to know why previous work (https://arxiv.org/abs/2203.10675) has reported gains while going in the opposite direction (i.e., removing gender information). A comparison with such line of research would be very interesting.
 3. Lack of statistical significance testing. Although the gaps between methods are large, confidence intervals of paired statistical significance would give more information when results are quite close (e.g., line 2 and 4 of table 3, ...).
 4. The effect of the solution on sentences where there is more than one referent is not evaluated.

**Reproducibility:**

3: Could reproduce the results with some difficulty. The settings of parameters are underspecified or subjectively determined; the training/evaluation data are not widely available.

**Reviewer Confidence:**

4: Quite sure. I tried to check the important points carefully. It's unlikely, though conceivable, that I missed something that should affect my ratings.

**Typos Grammar Style And Presentation Improvements:**

Line 388: "less" -> "lower"

In Table 1 up/down arrow for the metrics would help.

---

> ### Author Rebuttal · Authors · 2023-08-29
>
> Thank you for your positive review. We respond to your comments below.
>
> > 1. An impact statement is missing
>
> We fully agree with the need for an impact statement/ethical considerations due to the topic of our work and we will update our paper to include it. Please see our response on the ethics review for more detail.
>
> > 2. It would be interesting to know why previous work has reported gains while going in the opposite direction
>
> Thank you for pointing out a relevant and interesting work of (Fleisig and Fellbaum, 2022). We believe that while previous work’s approach seems to be in the opposite direction of ours, the motivation itself is shared in that both works try to “fix” model representations from having a fixed, stereotypical gender information. Their approach addresses this by removing gender information from encoder output adversarially when the given input has no gender context (via masking the pronouns in the input; see Figure 2 of their paper). Our approach, on the other hand, addresses this by injecting correct contextual gender information from encoder output contrastively when given inputs have gender contexts. We will update our paper to include the related work.
>
> > 3. Lack of statistical significance testing
>
> Thank you for the suggestion. We share the results on paired randomized permutation tests (with number of resamples N=100,000) for comparing line 2 (GACL) and line 4 (Ablation with GC loss only) of table 3. We found that the GACL method shows higher accuracy than the ablation on 10 out of 10 evaluated languages on WinoMT dataset and 4 out of 8 evaluated languages on MT-GenEval dataset directions with statistical significance of p-value less than 0.05. We will include the specific p-values as well as individual results for each evaluated target languages in the paper.
>
> > 4. The effect of the solution on sentences where there is more than one referent is not evaluated.
>
> We decided to focus on sentences with single referents in our work as 1) we found that existing NMT models still exhibit considerable stereotypical gender biases even with single referents, 2) there are statistically fewer cases compared to sentences with single referents (ratio of 7.2:1 on single to multiple referents during our filtering stage), and finally 3) lack of evaluation data/metrics considering gender bias of multiple referents. However, we believe this direction would be the natural next step for our work along with handling gender-ambiguous entities.
>
> > Q. In lines 127-131 it is stated that the training data is created so that it is balanced. Is this a requirement for the method? Has it been tested with unbalanced data and resampling? Was the method tested in the context of ambiguous referents?
>
> We require sampling the same number of masculine and feminine samples in a single mini-batch during fine-tuning so that the number of positive and negative pairs for both genders are equal. While not tested, we believe our method can also be fine-tuned with unbalanced data as long as balanced resampling is applied. Our method was not tested in the context of ambiguous referents.
>
> Overall, thank you for your comments. We will also fix the typos and include the missing references.

---

### Official Review · Reviewer_fQoU · 2023-08-04

**Soundness:** 3

**Excitement:**

4: Strong: This paper deepens the understanding of some phenomenon or lowers the barriers to an existing research direction.

**Paper Topic And Main Contributions:**

This paper attempts to mitigate gender bias in multilingual MT models by fine-tuning the encoder with a "gender-aware contrastive loss" in addition to the usual log likelihood. They also regulate with a teacher-loss relative to the original model.

**Questions For The Authors:**

While space is limited in this version, I would be interested if some qualitative examples could be given? Both where it does and does not work.

**Reasons To Accept:**

- This is an interesting and possibly novel approach to bias in MT. It seems straightforward and sensible, and is clearly explained.

- There are a good set of experiments, including a very wide range of languages, multiple different pre-trained models, as well as implementations of a sensible selection of similar approaches from the literature, with evaluation on two gender test sets with multiple metrics, and on Flores. I liked also the ablation experiments on using the different elements of GACL in isolation.

- The results seem good overall, and their analysis is very interesting, particularly the results for the effect on language directions which were not directly trained.

**Reasons To Reject:**

- The data selection process, which seems critical to the performance of the approach, is underspecified. It is briefly described in section 2.1, but important details are unclear: How does balanced sampling take place, are one or both of the M/F sentence sets undersampled? How is the domain difference between sentences with masculine vs feminine terms handled?

- The finetuning hyperparameters are a little surprising - they don’t seem equally appropriate for a very tiny handcrafted set of 388 segments and a reinterpretation of the Choubey et al gender dataset which in that paper ran to hundreds of thousands of segments. Does it make equal sense to fine-tune on both for a “few thousand steps”?



- The assumption is not unique to this paper, but I would strongly dispute that WinoMT contains examples of "unambiguous cases where there is only one correct translation with respect to gender". Firstly, it is possible to reinterpret many examples such that the pronoun is coreferent with either or neither of the professional entities in the input. Secondly, there are two professional entities in each sentences, and the "correct" translation of only one is implied by the pronoun - how is the other translated before or after the intervention? These are issues that have been raised in the literature for a while (Saunders et al 2020 NMT doesn't translate gender coreference right unless you make it, Gonzalez et al 2020 Type B Reflexivization as an Unambiguous Testbed for Multilingual Multi-Task Gender Bias) and the paper would be stronger if it addressed how they interact with the proposed approach.

- The approach seems to hinge on a concept of binary gender - aiming for a 50/50 likelihood of the correct vs incorrect gender via the contrastive loss. It would strengthen the paper to discuss how such an approach might be extended for, for example, neutralization.


Edit to add: The authors have responded clarifying many of these points. I am overall positive about this paper and would be happy to see it accepted with the proposed additional discussion and clarification.

**Reproducibility:**

3: Could reproduce the results with some difficulty. The settings of parameters are underspecified or subjectively determined; the training/evaluation data are not widely available.

**Reviewer Confidence:**

4: Quite sure. I tried to check the important points carefully. It's unlikely, though conceivable, that I missed something that should affect my ratings.

---

> ### Author Rebuttal · Authors · 2023-08-29
>
> Thank you for your detailed review and constructive comments. We respond to them below.
>
> * On the data selection process:
>
> Similarly to the GFST approach by Choubey et al., we undersample the larger of the M/F sentence sets to match the size of the smaller one. For the WMT18 en-de dataset we used, 415,401 masculine and 86,431 feminine sentence sets are extracted, and masculine sentence sets are undersampled with random sampling while all samples from feminine set are used to create a final balanced set of total 2*86,431=172,862 samples. We do not consider or perform additional processing to handle domain differences. We will update our paper to include these details.
>
>
> * On the finetuning hyperparameters:
>
> We first like to clarify that the GFST baseline was trained for 1,900 steps, the Handcrafted baseline was trained for 200 steps, and GACL method was trained for 700 steps (Table 5 in Appendix), and the number of these fine-tuning steps were not fixed but rather based on early stopping on the best MT-Gender dev set performance (Line 288). Choubey et al may have required 30K steps for training their en-de model (Appendix A of their paper) because in their work they trained the transformer model from scratch. Our implementation, on the other hand, fine-tunes an already pre-trained multilingual NMT model and was able to converge in a fewer number of steps.
> We will update the wording of “few thousand steps” (Line 376) to be more specific in our paper.
>
> * On the WinoMT dataset:
>
> Firstly, we agree that the WinoMT dataset is not ideal and some examples are not perfectly unambiguous in its interpretation. Consequently, we included evaluation on a more recent dataset, MT-GenEval, in our paper, which consists of professionally translated sentences containing unambiguous references to a single gender. With MT-GenEval, we show consistent findings that existing NMT models exhibit stereotypical gender bias and our approach helps mitigate the issue.
>
> Secondly, on the translation of the other secondary entity in the WinoMT which is (supposedly) ambiguous, we have run additional experiments using the variation dataset from (Saunders et al., 2020) and share the results. We found that all previous methods as well as ours lead to an consistent increase in gender accuracy of secondary entity by about 5% compared to gender accuracy of primary entity (Baseline: 46.3/48.6, GFST: 47.5/52.2, Handcrafted: 55.2/59.0, GACL 67.4/72.7), which is consistent with findings by (Saunders et al., 2020). We believe additional techniques (such as tagging, proposed by Saunders et al., 2020) to address this trend and possible ways handle the gender of an ambiguous entity is a great topic for future research, but out of scope of our work. We will update the paper to include the experimental results and this discussions.
>
>
> * On the discussion of extending beyond binary gender:
>
> Our work evaluated bias mitigation for the binary gender, which is a limitation that we will also clearly state in our ethical considerations section (please see response to ethics review). However, we believe our approach has a great flexibility to be extended to cover non-binary genders or gender-neutralizing translations. The supervised contrastive loss we employed for gender contrastive loss is not limited to binary classification tasks but applicable to multi-class labels, as the formulation relies only on defining positive pairs as samples from the same class, and negative pairs as samples from any different class. Correspondingly, augmenting the translation data with gender-neutral/inclusive versions of the translations and assigning pseudo-labels to apply multi-class supervised contrastive loss is viable within our framework. We will include these discussions in our paper.
>
>
> * On qualitative examples:
>
> Thank you for the suggestion. We will include examples of translations before and after applying our fine-tuning method in the appendix of our paper.
>
> Overall, thank you for your comments. I'll be happy to engage in any further discussion.

---

### Meta-Review · Area_Chair_24Tx · 2023-09-15

**Recommendation:** 5

**Metareview:**

This paper proposes reducing gender bias in multilingual MT models by fine-tuning the encoder with an additional "gender-aware contrastive loss".

The results on different languages show that the approach reduces gender bias while retaining translation quality.

The analysis explaining the strengths of the proposed approach is also presented.



Revisions discussed during the authors' response should be applied.

Also, dealing only with binary gender should be mentioned in limitations (while including non-binary it is very easy in English, for most of the other languages there are no clearly defined rules (yet)).

---

### Decision · Program_Chairs · 2023-10-07

**Decision:**

Accept-Main

**Comment:**

This paper proposes reducing gender bias in multilingual MT models by fine-tuning the encoder with an additional "gender-aware contrastive loss".

The results on different languages show that the approach reduces gender bias while retaining translation quality.

The analysis explaining the strengths of the proposed approach is also presented.



Revisions discussed during the authors' response should be applied.

Also, dealing only with binary gender should be mentioned in limitations (while including non-binary it is very easy in English, for most of the other languages there are no clearly defined rules (yet)).